# Cortical Transformers: Robustness and Model Compression with Multi-Scale Connectivity Properties of the Neocortex.

**Brian S. Robinson**[1] **and Nathan Drenkow**[1,2]

[1]Johns Hopkins University Applied Physics Laboratory
[2]Johns Hopkins University
`{brian.robinson, nathan.drenkow}@jhuapl.edu`

## Abstract

Transformer architectures in deep learning are increasingly relied on across domains with impressive results, but the observed growth of model parameters may be unsustainable and failures in robustness limit application. Tasks that are targeted across domains by transformers are enabled in biology by the mammalian neocortex, yet there is no clear understanding of the relationship between processing in the neocortex and the transformer architecture. While the relationship between convolutional neural networks (CNNs) and the cortex has been studied, transformers have more complex computations and multi-scale organization, offering a richer foundation for analysis and co-inspiration. We introduce a framework for enabling details of cortical connectivity at multiple organizational scales (micro-, meso-, and macro-) to be related to transformer processing, and investigate how cortical connectivity principles affect performance, using the CIFAR-10-C computer vision robustness benchmark task. Overall, we demonstrate the efficacy of our framework and find that incorporating components of cortical connectivity at multiple scales can reduce learnable attention parameters by over an order of magnitude, while being more robust against the most challenging examples in computer vision tasks. The cortical transformer framework and design changes we investigate are generalizable across domains, may inform the development of more efficient/robust attention-based systems, and further an understanding of the relationship between cortical and transformer processing.

## 1 Introduction

Transformers [30] have enabled many recent successes, especially in computer vision [8; 20], but open challenges remain for increasing robustness and reducing model size [15]. Tasks across domains typically targeted by transformers are enabled by the mammalian neocortex [24], which achieves efficient processing and real-world robustness. While connections between transformer networks and principles of processing in biological networks have been made (e.g. between models and encoding in the hippocampal formation [31] and between principles of attractor dynamics and sparse reconstruction [27]), it is unknown how to leverage cortical connectivity properties explicitly into transformer networks. Relative to convolutional neural networks (CNNs) which have been studied [19; 17; 18], transformer networks offer many potential advantages in regards to connections to cortical processing yet remain under-explored. In particular, transformers 1) have a more complex multi-scale organization, with attention being calculated at the smallest scale as in the cortical microcircuit 2) enable the seamless incorporation of recurrence and feedback within and across levels

4th Workshop on Shared Visual Representations in Human and Machine Visual Intelligence (SVRHM) at the Neural Information Processing Systems (NeurIPS) conference 2022. New Orleans.

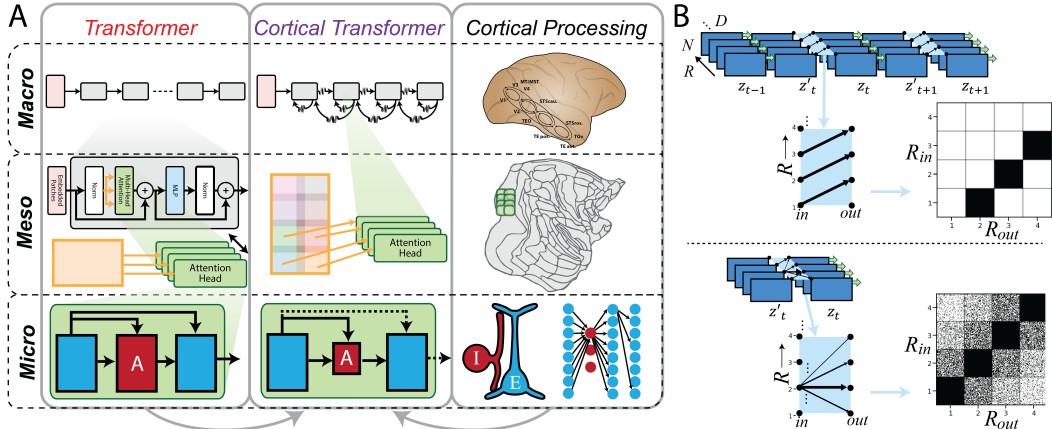

Figure 1: Incorporating multi-scale connectivity properties of the cortex. A. Macro-scale: 1) recurrent processing between cortical blocks, 2) sparsity, between-block connections have exponentially increasing levels of sparsity, 3) token-interactions (twisted arrows), learnable parameters are introduced for how tokens can interact to break the fixed spatial correspondence through subsequent layers. Meso-scale: model embeddings are parallelized between attention heads, with partial overlap based on a 2D structure. Micro-scale: attention-embeddings (A) are based on inhibitory (I) neurons with reduced population size (dimensionality-reduction) and denser connections , while forward-embeddings (blue) are based on excitatory (E) neurons (sparser connectivity). B. Recurrent processing of the model's latent space $z_t$ ($R$ regions, $N$ tokens, $D$ dimensionality per region), can support feedforward region connectivity (top), or support more complex region-to-region connectivity (bottom). Note: Macro-/Micro-Cortical Processing figure components adapted from [16] and [29] respectively.

of processing, and 3) naturally support multi-modal processing. Additionally, while CNNs embody explicit inductive biases based on observed hierarchical properties of cortical neural responses, attention-based networks implicitly instantiate, replace, and/or expand on these biases [5; 25; 14]. In this work, we introduce a framework for integrating a series of structural cortical properties into transformer networks which enables both a method to study the computational effects of these properties as well as a way to augment modern deep learning architectures.

To investigate details of cortical processing for transformer networks, we focus across macro-, meso-, and micro-scales. At the **macro-scale**, we make the comparison between cortical areas and successive layers in a transformer network, extending the correspondences between areas and layers utilized in CNN models of cortical processing [33; 19; 17]. In this work, we focus on three macro-scale connectivity principles: 1) *recurrence*: Standard transformer networks use purely feedforward layers, which is starkly different than the recurrent and feedback processing that is observed between cortical areas [16]; 2) *sparsity*: Cortico-cortico connections between areas at varying hierarchical levels become exponentially more sparse as distance between areas increases [22]; 3) *token-interactions*: Unlike the cortex where visual receptive fields vary considerably in size between areas [26], standard transformer networks in vision [8] rely on a fixed set of patch-wise feature tokens with specific spatial correspondences which are preserved throughout subsequent layers of processing.

At the **micro-scale**, we relate cortical micro-circuity properties [11] of excitatory neurons (which have longer-range output connectivity) and inhibitory [29] neurons (which have shorter-range connectivity) to the transformer attention head calculation. The computational role of inhibitory neurons can be complex, especially in the case where inhibitory neurons, that synapse onto dendritic regions (e.g., Sst neurons), non-linearly and multiplicatively interact with patterns of excitatory input [28]. In the transformer's attention head, the attention-embeddings ($Q, K$) utilize non-linear, multiplicative interactions whereas forward-embeddings ($V$) propagate only to downstream layers. We inform the structure of these calculations with inhibitory and excitatory neuron population properties respectively. In this work, we focus on two micro-scale connectivity principles: 1) dimensionality-reduction, where excitatory neurons outnumber inhibitory neurons; 2) sparsity, where excitatory-excitatory connections have greater sparsity than inhibitory-excitatory connections [10].

Between the micro-/macro-scales lies a **meso-scale**, where neurons lie in the same area of the cortical sheet, but still exhibit distance-dependent connectivity [22], which we implement by imposing a 2D partially overlapping input to each attention head.

For experiments, while transformers have already shown impressive performance on standard vision benchmarks [8; 32; 3], we consider dimensions of performance where biological vision is still dominant. We examine the challenge of robustness to natural corruptions where transformers have demonstrated improvements over CNNs [2; 23] yet the robustness problem remains far from solved [9].

In this work, we first introduce the Cortical Transformer architecture which incorporates several multi-scale cortical connectivity properties across macro-, meso-, and micro-scales. In a series of computational experiments using the CIFAR-10-C [13] corruption benchmark, we demonstrate the efficacy of our model and investigate how these cortical properties affect robustness and model compression. We find that incorporating the investigated cortical properties improves performance on the most difficult image corruptions for baseline transformer networks while reducing model size.

## 2 Methods

To study the properties of cortical connectivity, we introduce a series of generalizations of the transformer networks across macro, micro, and meso scales which comprise a single Cortical Block. Stacking these Cortical Blocks depth-wise, produces our Cortical Transformer (CT).

**Macro** At the macro-level, we introduce a recurrence-enabled generalization of the standard feedforward transformer, where the latent space, $z_t \in \mathbb{R}^{N \times D \times R}$ is divided into $R$ regions and processed recursively for each timestep $t$ (with $N$ tokens and $D$ dimensionality per region). First, a multi-region self-attention (MRSA) is calculated, which is a generalization of multi-headed self attention (MHSA),

$$z'_t = MRSA(RN(z_{t-1})) + z_{t-1}, \tag{1}$$

where MHSA is first calculated separately for the latent space in each region, $z_r \in \mathbb{R}^{N \times D}$, then concatenated across regions, followed by normalizing using Region Normalization (RN) as further specified in the Appendix A. Next, a multi-region perceptron (MRP) is calculated, which is a generalization of the multi-layer perceptron that allows interactions between regions,

$$z(t) = MRP(RN(z'_t)), \quad MRP(z) = \sum_{r=1}^{R} u(z_r B_r) + u(W^{M,r} g(z_r W^{1,r}) W^{2,r}) \tag{2}$$

The $MRP(.)$ calculation is performed in a parallel fashion for the latent space in each region, with:

- learnable parameters for MLP weights ($W^{1,r} \in \mathbb{R}^{D \times D_{FF}}$, $W^{2,r} \in \mathbb{R}^{D_{FF} \times D}$) and $W^{M,r} \in \mathbb{R}^{NR \times N}$ enabling interactions between tokens in different regions,
- $B_r \in \mathbb{R}^{D \times DR}$, a binary weight matrix that controls how the residuals propagate,
- $g(.)$, a GeLU nonlinearity, and
- $u(.)$, an unstacking/reshaping operation of a matrix with a total of $NDR$ elements to a tensor in $\in \mathbb{R}^{N \times D \times R}$ (applied to $z_r B_r \in \mathbb{R}^{N \times DR}$ and $W^{M,r} g(z_r W^{1,r}) W^{2,r} \in \mathbb{R}^{NR \times D}$).

When $B_r$ and $W^{M,r}$ are frozen with identity matrix configurations further specified in Appendix A, (1) and (2) simplify to a standard feedforward transformer architecture with $L$ layers when the total number or timesteps and number of regions both equal $L$. Alternatively, $B_r$ and $W^{M,r}$ can be structured to enable different forms of region-to-region interactions and modeled on observed exponential drop-off cortical connectivity (See Appendix A). Furthermore, the number of timesteps can be increased to enable further levels of processing with a fixed number of parameters.

**Micro** As further outlined in Appendix A, the self-attention calculation is parameterized by $W^Q \in \mathbb{R}^{D \times D_q}$, $W^K \in \mathbb{R}^{D \times D_q}$, and $W^V \in \mathbb{R}^{D \times D_v}$ for each of the $H$ attention heads for each region and $W^0 \in \mathbb{R}^{D_v H \times D}$ for each region. These parameters can be sorted by use as either attention-embeddings ($W^Q$ and $W^K$) or forward-embeddings ($W^V$ and $W^0$). Dimensionality-reduction may be applied by reducing $D_q$ relative to $D_v$ by a factor of $d$ and sparsity may be applied by initializing each element of $W^V$ and $W^0$ to be sparse with probability $s$.

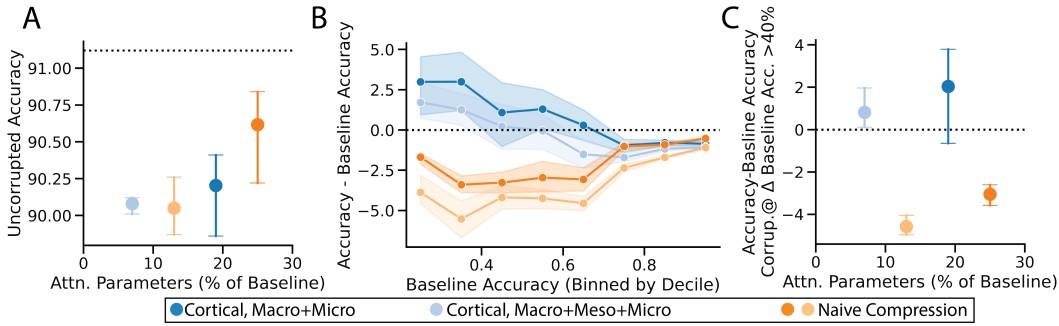

Figure 2: CIFAR-10-C accuracy and model compression with multi-scale cortical connectivity properties. A. Model accuracy on uncorrupted CIFAR-10 is slightly reduced for cortical and naive compression network variants. B. Model accuracy (relative to baseline) versus corruption difficulty. For more difficult corruptions, cortical network variants have increased accuracy over the baseline network and naive compression variants. The 71 evaluated corruptions are binned by decile of baseline accuracy. C. Average model accuracy on the subset of the most difficult corruptions for the baseline model (corruptions that reduced baseline model accuracy by over 40%). Dotted lines signify baseline network averages. All comparisons are run with 3 random initialization seeds. Error bars in A and C are min and max values, error bands in B are bootstrapped 95% confidence intervals.

**Meso** At the meso-scale we constrain the input dimensionality ($D$) of each attention head separately. We decompose each of the $W^{E,h}, E \in \{Q, K, V\}, h \in \{1...H\}$ attention embedding matrices, $W^{E,h} = S^h \tilde{W}^{E,h}$ using a smaller set of learnable weights ($\tilde{W}^{E,h}$). The binary matrix $S^h \in \mathbb{R}^{D \times D_s}$, ($D_s < D$) has at most one non-zero element in each row so that that each head receives input from a different subset of elements along the dimensionality, $D$. The specification of $S^h$ can enable completely parallel inputs to each attention head or 1D and 2D patterns of overlap as further described in Appendix A.

## 3 Experiments and Results

To investigate properties of cortical processing in transformer networks, we perform a series of computational experiments using the CIFAR-10-C benchmark. We utilize CCT [12] as a baseline network given its parameter efficiency and competitive performance on the standard CIFAR-10 benchmark compared to other vision transformers. Importantly, the CCT network preserves a standard transformer-based backbone, obtaining parameter efficiency and performance gains through an updated image tokenizer before the transformer-based backbone and a sequence pooling operation for classification after the transformer-based backbone. We leverage the convolutional tokenizer from this architecture and replace the transformer backbone with our Cortical Blocks to evaluate cortical design principles. Additional model and training details are found in Appendix B.1 and all code will be released at `https://github.com/aplbrain/cortical_transformers_22`. For both the CCT baseline and our Cortical Transformer, a baseline depth of four transformer layers/Cortical Blocks respectively is used, consistent with the four cortical regions associated with visual object recognition in the primate ventral stream [7].

**Setup:** For a general comparison of robustness performance and network size, we train networks with all cortical design principles on CIFAR-10 and evaluate them on CIFAR-10-C. We compare networks with macro+micro+meso and macro+micro cortical variations to a baseline transformer network as well as networks that achieve similar model compression with naive downscaling approaches (Fig. 2). For measuring robustness to corruptions, we quantify the difficulty of a corruption type and severity level as the baseline transformer performance (considering that corruption type has a considerable influence on the practical difficulty of a task in addition to the benchmark annotated severity level).

**Macro-, Meso-, and Micro-Analysis** We first examine how combinations of cortical principles (outlined in Section 2) impact performance. In aggregate, these cortical variations (which have a $5\times$ and $15\times$ compression in attention parameters) yield minor decreases in uncorrupted image accuracy, but as the corruption difficulty increases, the cortical variations have higher relative performance (Fig. 3). Naively reducing attention parameters by reducing the dimensionality of

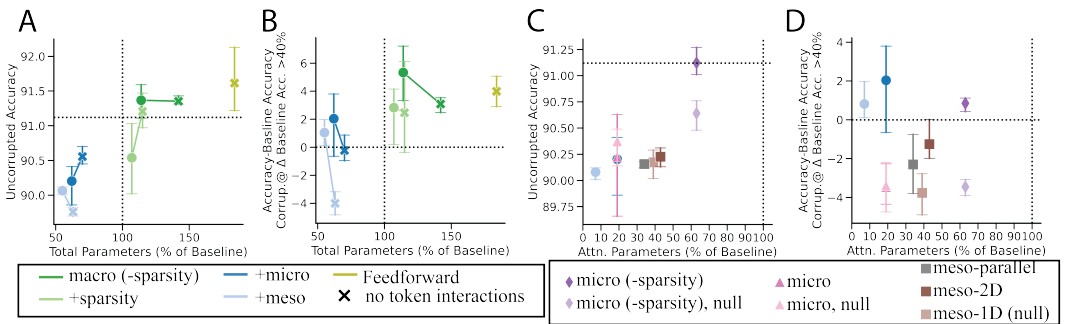

Figure 3: Effect of individual connectivity properties on CIFAR-10-C accuracy and model compression. A-B. Macro analysis, recurrence (-sparsity) increases all performance, token-interactions enhance performance on difficult corruptions. C-D. Micro and Meso analysis, all micro and meso connectivity principles reduce learnable attention parameters. While there is some trade-off in performance for most micro and meso network variants, the micro variant with dimensionality reduction only (-sparsity) has increased performance. In comparison to null non-cortical control network variants, the micro variant with dimensionality reduction only (-sparsity) and the meso-2D network variants have increased performance. (A, C). Model accuracy on uncorrupted CIFAR-10. (B,D). Average model accuracy on the subset of the most difficult CIFAR-10-C corruptions for the baseline model (corruptions that reduced the baseline model accuracy by over 40%). Error bars signify min/max values for three seeds. Dotted lines signify baseline network averages. Feedforward macro results are with the same number of processing steps (8).

$W^Q, W^K, W^V, W^0$ in standard transformers by similar factors (reducing both $D_v$ an $D_q$ by a factor of 4 or 8), fails to improve accuracy for both the uncorrupted and most difficult corruptions. Thus, using the Cortical Transformer enables the number of attention parameters to be reduced by over an order of magnitude while having increased performance on the most difficult corruptions.

**Ablation studies** We perform additional ablation studies to quantify the relative contributions of the macro-/meso-/micro- design principles investigated (Fig. 3). At the macro-level, investigated principles include recurrence, sparsity, and token-interactions. Adding recurrence is the only macro-principle investigated that increases performance across the board for uncorrupted and the most difficult corruptions (and even has better performance on difficult corruptions than baseline models with twice the depth). Intuitively, recurrence may be beneficial by increasing the virtual depth of the network via additional processing steps but without increasing parameters. Token-interactions improve average accuracy on difficult corruptions across all tested comparisons (with the strongest gains seen in the most compressed networks). A potential interpretation for the benefit of token-interactions for difficult corruptions is that expected visual features in heavily corrupted data may be imputed better by breaking the fixed spatial correspondence per token and adding computations that enable additional interactions between spatial features. Adding sparsity does reduce model parameters, but at the expense of decreasing relative performance which is in line with a general expected tradeoff between model size and performance.

The investigated micro- and meso- connectivity principles can be further examined both in isolation and relative to null non-cortical variants that can further probe the benefits of the investigated cortical principles. The investigated micro- connectivity principles include dimensionality reduction and sparsity (see Section 2). The dimensionality reduction enhanced the performance on difficult corruptions, and was the only investigated parameter-reducing principle which enhanced performance, bucking an expected tradeoff between model size and performance. With both sparsity and dimensionality reduction, the total number of attention parameters can be reduced further to a factor of 5, but with exhibiting some trade-off in model size and performance. We test a null non-cortical control by switching the parameters for dimensionality-reduction ($\{W^Q, W^K\}$ to $\{W^V, W^0\}$), and find a marked drop in performance. A similar null non-cortical control is also introduced for sparsity ($\{W^V, W^0\}$ to $\{W^Q, W^K\}$), with a minimal difference in performance, further highlighting the relative contribution of the dimensionality-reduction as a micro- connectivity principle.

The investigated meso- cortical connectivity principle with a 2D overlap in input dimensionality has two general properties: 1) the input to each attention head is not the full model dimensionality and 2) the organization of the overlap to each attention head follows a two dimensional pattern of overlap

(Fig. 3C-D). To further understand these properties, two null comparison are run: 1) where input dimensionality to each attention head is completely parallelized and 2) where the overlap of input dimensionality has a 1D instead of a 2D overlap. When the input to each attention head is completely parallelized, there is an expected trade-off in model size and performance. When the overlap of input dimensionality is 2D vs. 1D, however, the 2D overlap has a greater accuracy on difficult corruptions, suggesting the importance of the 2D organization of overlap in dimensionality.

## 4  Discussion

In this work, we introduce a framework for relating cortical to transformer processing at multiple organizational scales: 1) (macro) between area-to-between block, 2) (meso) within area-to-within block, and 3) (micro) microcircuitry properties-to-within attention head calculation. We introduce a set of principles across these scales and generalizations of the transformer architecture to support them. Using a CIFAR-10-C robustness benchmark, we found that these principles could reduce learnable attention parameters by over an order of magnitude while increasing the performance on the most difficult corruptions. In a series of ablation studies to better understand the relative importance of these principles, we found that the macro principles of recurrence and token interactions had the largest effects on increasing performance. The majority of the meso and micro principles exhibited a trade-off in performance when reducing model size (however with enhanced performance over null non-cortical variants). The micro dimensionality reduction (motivated by a reduction in the number of attention-calculating inhibitory neurons), however, enhanced model performance while reducing the number of parameters. When all investigated principle are combined, even though the observed performance increases were modest, the total number of learnable attention parameters are reduced by a factor of 15, which is a significant gain in model compression. Furthermore, the largest performance increases were observed for the most difficult natural corruptions, highlighting the potential utility of the investigated principles for domain shift in both biological and artificial systems.

For practitioners utilizing transformer networks, some of these investigated principles can be readily incorporated, such as micro-dimensionality-reduction or token-interactions, which had improvement in model compression and robustness respectively. Several of these principles are also related to existing design augmentations for transformer networks. For example, macro-sparsity is related to sparse transformer implementations [4] (which are applied to the token dimension as opposed to the model dimension). Additionally, the macro-recurrence extends recurrence approaches applied to encoder-decoder language processing models [6]. Finally, the learned token-interactions introduced here are a flexible method of enabling hierarchical token representation, related to more rigid patch merging operations utilized in hierarchical vision transformers [20].

Our work also extends research into the relationship between visual cortical processing and convolution-based networks [19; 17] by 1) adding the ability to investigate the relationship at multiple spatial scales (including the calculation of attention in cortical microcircuity) and 2) adding the ability to make comparisons with networks utilized across domains, such as language. To gain a more thorough understanding of the investigated connectivity principles, it will be critical to evaluate them on a wider range of tasks. Further investigation into the relationship between cortical processing across scales and transformer networks could also lead to additional experimental comparisons, for example by comparing properties of inhibitory neural responses to calculated attention masks. Overall, the framework, multi-scale connectivity principles, and computational experiments presented in this work have considerable room for further investigation to enable more compressed/robust transformer networks and to further an understanding of the relationship between cortical processing and modern deep learning approaches.

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

## Acknowledgments and Disclosure of Funding

The work received funding from an internal grant from the Johns Hopkins University Applied Physics Laboratory. We would like to acknowledge Brock Wester for help in creating the graphics, as well as Michael Wolmetz, Brock Wester, and Erik Johnson for helpful feedback on this work.


# A  Cortical Block

## A.1  Macro Details

**Multi-region self attention**  For the latent space in each region ($z_r \in \mathbb{R}^{N \times D}$), self-attention (SA) and multi-headed self-attention (MHSA) is calculated as in standard transformer networks [30] across all $H$ attention heads,

$$q = z_r W^Q, k = z_r W^K, v = z_r W^V, A = softmax(qk^T/\sqrt{D}), SA(z_r) = Av \qquad (3)$$

$$MHSA(z_r) = [SA_1(z_r); SA_2(z_r); ...; SA_H(z_r)]W_r^0 \qquad (4)$$

For multi-region self-attention ($MRSA$), the MHSA is concatenated between all regions,

$$MRSA(z) = Concat(MHSA(z_{r=1}), MHSA(z_{r=2}), ..., MHSA(z_{r=R})). \qquad (5)$$

**Multi-region perceptron**  In Eq. 2, which specificies the calculation of the multi-region perceptron, there are learnable output token-interactions parameters for each regions, $W^{M,r} \in \mathbb{R}^{NR \times N}$. There is a block structure to $W^{M,r}$, which is created by stacking the effective token-interactions for each output region ($W^{r_{out}=1}, W^{r_{out}=2}, ..., W^{r_{out}=R}$), where $W^{r_{out}} \in \mathbb{R}^{N \times N}$. To enable a standard feedforward architecture, the block in $W^{r_{out}}$ where $r_{out} = r + 1$ is an identity matrix and the rest of $W^{M,r}$ is all zero. To enable fully recurrent interactions between regions, $W^{M,r}$ is fully learnable and non-zero. To model exponential drop-off in connectivity between regions, each element in $W^{r_{out}}$ is masked to zero with a probability $p_m = 1 - e^{(|r-r_{out}|)/\lambda}$, where $\lambda$ controls the rate of exponential connectivity drop-off.

Similar to specification of $W^{M,r}$, configurations of residual propagation can be enabled with $B_r \in \mathbb{R}^{D \times DR}$, composed of stacked ($B^{r_{out}=1}, B^{r_{out}=2}, ..., B^{r_{out}=R}$) blocks, where $B^{r_{out}} \in \mathbb{R}^{D \times D}$. To enable a standard feedforward architecture, $B^{r_{out}}$ is an identity matrix at $r_{out} = r + 1$ and zero otherwise. To enable fully recurrent interactions, all $B^{r_{out}}$ blocks are an identity matrix. To model differences in connectivity between regions, the same calculation of $p_m$ above is used to control the probability of masking elements to zero.

An additional variation of the multi-region perceptron can be formulated that enables region to region connectivity specification, but without token-interactions,

$$M\tilde{R}P(z) = \sum_{r=1}^{R} u(z_r B_r) + u(g(z_r W^{1,r})W^{\tilde{2},r}), \qquad (6)$$

where $W^{\tilde{2},r} \in \mathbb{R}^{N \times DR}$ is configured in a block structure with blocks $\tilde{W}^{r_{out}} \in \mathbb{R}^{N \times D}$. In this case, a standard feedforward architecture is specified by making $\tilde{W}^{r_{out}}$ where $r_{out} = r + 1$ learnable and $\tilde{W}^{r_{out}}$ equal to zero in other blocks.

## A.2  Meso details

When decomposing $W^{E,h} = S^h \tilde{W}^{j,h}$ for $E \in \{Q, K, V\}$, the learnable parameters, $\{\tilde{W}^Q \in \mathbb{R}^{D_s \times D_q}, \tilde{W}^K \in \mathbb{R}^{D_s \times D_q}, \tilde{W}^V \in \mathbb{R}^{D_s \times D_v}\}$ are lower dimensional than $W^{j,h}$ by a factor of $D_s/D$. When $D_s = D/H$, the input space can be completely parallelized between heads. When $D_s > D/H$, there can be overlap of input space between heads. To approximate a sheet-like two dimensional

organization of the input space between heads, the $D$ dimensions are reshaped to a two dimensional $D/m \times m$ matrix, M. The input of each attention head is parameterized with a center location $(x^h_{center}, y^h_{center})$ evenly distributed across M and a $width$, which controls the amount of input overlap between heads.

### A.3 Region Normalization

Just as LayerNorm [1] plays a key role in standard Transformer layers, we consider normalization within and across regions in the context of our Cortical Transformer. For input $x$ to the `RegionNorm` layer, we compute

$$z = \gamma \frac{x - \mu}{\sigma} + \beta \tag{7}$$

where $\gamma$ and $\beta$ are learnable parameters. Our `RegionNorm` differs from LayerNorm with respect to the calculation of $\mu, \sigma$ and the dimensions along which $\gamma, \beta$ are learned and applied. In particular, we allow the dimensions for computing $\mu, \sigma$ to be decoupled from the learnable paramter dimensions for $\gamma, \beta$. Our intent was to allow for variations where the normalizing factors could be global or local to regions (via the shape of $\mu, \sigma$) and the affine paramters could be global, by region, or elementwise (via the shape of $\gamma, \beta$).

In cases where $\mu, \sigma$ were computed globally, the regions become coupled. For instance, if activations in specific later regions are zero, then the activations in those regions will become non-zero (i.e., for region $i$, $z_i = \frac{-\mu}{\sigma}$ assuming $\gamma = 1, \beta = 0$ here). Our `RegionNorm` layer enables investigation of multiple interpretations of normalization in the context of cortical connectivity as they relate to within and across-region behavior. We explored many of these configurations, but for all experiments, we computed $\mu, \sigma$ along the token dimension only, and learn element-wise gains and biases ($\gamma, \beta$) for region and token dimensions.

## B  Experiment Details

### B.1  Baseline Architecture

We utilize the CCT architecture [12] both as a baseline network and a foundation for our cortical transformer model. A primary motivation of CCT was to reduce the trainable parameters by replacing the patch-wise tokenization in ViTs [8] with a convolutional tokenizer implemented as:

$$x_0 = MaxPool(ReLU(Conv2D(x))) \tag{8}$$

for input image $x$ and where $MaxPool, ReLU, Conv2D$ follow their standard implementations. The result is a decrease in the total number of parameters and an increase in the number of tokens. Following [12], we use the $3 \times 2$ tokenization which equates to two layers of $3 \times 3$ convolutional kernels with $3 \times 3$ (stride=1), and $3 \times 3$ pooling (stride=2). In the baseline and cortical transformer cases, convolutional layers were set to output feature maps with the channel dimension equal to 128. We utilize the Sequence Pooling operation from the original CCT formulation which uses attention-based pooling over the output tokens prior to a final classification layer. This considers the case that tokens may contain varying amounts of information and the learnable attention pooling parameters help to account for this variation.

### B.2  Training Hyperparameters

For training, hyperparameters are described in Table 1. We use cosine annealing [21] to adjust the learning rate over all epochs.

### B.3  Cortical Block Hyperparameters

**Baseline**  The baseline configuration for all variants have the number of heads per region, $H$=4, the dimensionality per region, $D = 128$, and number of regions, $R = 4$.

| Hyperparameter | Value |
|---|---|
| Base learning rate | 5e-4 |
| Batch size | 64 |
| Optimizer | AdamW |
| Weight decay | 3e-2 |
| Epochs | 200 |

Table 1: Training Hyperparameters

**Macro**   The rate of exponential drop-off, $\lambda = 0.5$.

**Micro** The investigated micro configurations dimensionality reduction by a factor of $d = 4$ and sparsity level, $s = 0.125$.

**Meso**   For the tested meso variants, $H$ is increased to 8 to investigate enhanced parallelization between heads. For specifying the investigated fully parallel configuration, $m = 8$ and $width = 4$. For the configuration investigating 2D overlap, $width = 6$. The null comparison with 1D input overlap is configured with $m = 1$, $width = 23$.

## C   Compute Resources

Experiments were run on an internal cluster containing NVIDIA v100 and a100 GPUs as well as additional g4dn.2xlarge AWS instances.

## D   Societal Impacts

We believe this work poses minimal societal risks as our primary objective is to make stronger connections between transformer architectures and cortical processing. We aim to improve model performance for challenging/corrupted inputs which should enable more positive uses of these models in real-world settings. Nonetheless, persisting failures of these models and/or their improved performance in the hands of bad actors could have negative societal impact.

