# OpenReview forum: "Cortical Transformers: Robustness and Model Compression with Multi-Scale Connectivity Properties of the Neocortex."
_NeurIPS.cc/2022/Workshop/SVRHM — SVRHM Poster_

### Official Review · Reviewer_cuH9 · 2022-10-14
**Technically solid paper that asks interesting questions**

**Rating:** 7
**Confidence:** 3

**Review:**

The paper presents a transformer architecture that the authors endow with certain connectivity properties at varying "scales" of computation to study robustness.

The paper is of high quality: well-written and professionally presented. The notation on page 3 is a bit dense, but I understand the space constraints.

While my familiarity with transformers is indeed limited, the work presents an exciting path forward towards understanding many properties that have been previously studied in a purely convolutional setting. The problem is at least well-motivated. However, to the uninitiated reader, the originality of the paper is somewhat unclear ("While some connections between transformer networks..." --->could you expand here how your model is different?). The review sets up the promise over previous studies with CNNs, but quickly jumps into the scale descriptions. Perhaps, for example, you could also motivate (in more text) how CCT is fundamentally different architecture.

The main results in figures 2 and 3 have very similar scales, especially for 3 seeds. Are the results statistically significant?

Also, for question 3(a), this should be answered "no," especially if the code isn't included with the submission. The justification provided is more in line with question 3(b).

---

> ### Author Response · Authors · 2022-12-28
> **Reviewer 4 Response**
>
> Again, we appreciate the detailed review and comments which have utilized to make updates to the paper:
>
> For, "The notation on page 3 is a bit dense, but I understand the space constraints.", we have updated the methods notation (and hope this makes the details easier to follow per reviewer 2's comment as well).
>
> For " ("While some connections between transformer networks..." --->could you expand here how your model is different?).", We have expanded our description to include "While connections between transformer networks and principles of processing in biological networks have been made (e.g. between models and encoding in the hippocampal formation (Whittington et al. 2021) and between principles of attractor dynamics and sparse reconstruction (Shi et al. 2022)), it is unknown how to leverage cortical connectivity properties explicitly into transformer networks."
>
> For "The review sets up the promise over previous studies with CNNs, but quickly jumps into the scale descriptions", we added an additional segue sentence at the end of the introduction to contextualize the scale descriptions.
>
> For "Are the results statistically significant?", as we have mentioned in the response to reviewer 3, we did not have the computational resources to run enough trials to enable more traditional statistical comparison techniques, however we reported the full min-max range of results in all quantitative comparisons to give the reader a conservative estimate of our uncertainty in the reported trends.
>
> For "question 3(a), this should be answered "no," especially if the code isn't included with the submission. The justification provided is more in line with question 3(b).", We have updated the methods and question 3(a) with the hyperlink where all code will be released.

---

### Official Review · Reviewer_PL8e · 2022-10-14
**Promising results showcasing the potential benefits of cortical based architectures in transformers**

**Rating:** 7
**Confidence:** 3

**Review:**

Summary: In this study, three cortical mechanisms (macro, meso, and micro) were added to a transformer model and the accuracy was investigated for uncorrupted and non-adversarial images from the CIFAR-10 dataset. Overall, the models showed a similar if not slightly better performance on corrupted images but with fewer attention parameters than a typical transformer. The reduction in accuracy on the uncorrupted dataset appears to be negligible as well.

Strong Points:
1. The results clearly show how a variety of cortical transformers perform comparably or slightly better than a typical transformer (especially on the most difficult corrupted images) despite having fewer attention parameters.
2. The biological inspiration of the cortical mechanisms across a range of scales is well aligned with the workshop theme and showcases the potential benefits of recurrences and overlapping regions.

Suggestions:
1. Consider running more trials to have greater statistical certainty of the results (standard deviations in accuracies)
2. Figure 3 is a bit cluttered and confusing with all the different model types. I think it would be clearer, especially for colorblind individuals, if different shapes were used for the different models or a different color scheme were used that is more colorblind-friendly.
3. Figure 2 would be a lot clearer if the naïve compression model were explained in the methods section.
4. While the results on the non-adversarial network were well presented, I am curious how the cortical transformer models would perform on adversarial examples.
5. I wish there was more of a discussion about why certain models performed better than others. For example, the macro+micro performed the best but adding the meso mechanisms seemed to decrease the accuracy. Why do you think this might be the case?

---

> ### Author Response · Authors · 2022-12-28
> **Reviewer 3 Response**
>
> Thank you for the detailed review and helpful suggestions, which we comment on in order below:
>
> 1) This is an important caveat for our results. Enhancing beyond three trials per network configuration would enhance the confidence of trends observed in our results.  In this work, however, we were limited in the number of trials we could run based on our available computational resources, and thus reported the full min-max range of results in all quantitative comparisons to give the reader a conservative estimate of our uncertainty in the reported trends.
>
> 2) We have updated the colors in Fig 3. A/B and marker shapes in Fig 3. C/D to make more color-blind friendly.
>
> 3) We have updated the description of naive compression in the results section to reference the specific dimensions of the methods that are reduced "(reducing both $D_v$ an $D_q$ by a factor of 4 or 8)".
>
> 4) We agree that evaluating on a more diverse set of tasks would be beneficial for characterizing the properties and that the performance on adversarial tasks could be important to contextualize our results in the broader field. We chose to focus on natural corruptions (non-adversarial robustness) as these are a type of corruption expected to be seen by biological systems and a type of corruption where artificial systems are still lacking. We have updated the discussion section to highlight the need to extend our evaluation to additional tasks and have updated our introduction to call out that we investigate the performance on "natural corruptions".
>
> 5) We have augmented our results (revamped last 3 paragraphs) and discussion (added 5 sentences to the first paragraph) to highlight why certain models perform better than others.

---

### Official Review · Reviewer_bRig · 2022-10-14
**Potentially very relevant innovation to reduce number of learnable attention parameters in transformer networks**

**Rating:** 7
**Confidence:** 2

**Review:**

The authors employ an architecture, with principles at three different levels or organisation, and evaluating how well their network performance on cifar 10 (also with degration on cifar 10).
They show that this network specifically (and also all 3 components necessary) perform better than naive compression.
I have trouble following quite a lot of details so am unsure of my review. It might be some of the text where to much happens without a lot of detail but it is also me.

Reducing the number of learnable parameters would make transformers much more digestible as a proposed mechanisms for human processing.
However, cifar 10 is two simple to make any point definative.

---

> ### Author Response · Authors · 2022-12-28
> **Reviewer 2 Response**
>
> Again, we appreciate the feedback. The concern of the difficulty of following the details was also mentioned by reviewer 4, who in particular mentioned that the notation on page 3 (where the methods are presented) was quite dense. We have updated the methods notation with further/less dense description and hope that this will help readers follow the method details more easily.
>
> Additionally, the comment that utilizing CIFAR 10, a relatively simple computer vision task, being too simple to make the points definitive, is an important limitation of this current work. We have updated the discussion section to explicitly call this limitation out to the reader. Nevertheless, we feel that the results obtained on CIFAR 10 had consistent trends that are still informative for the community and that can be investigated in future work.

---

### Official Review · Reviewer_5hGV · 2022-10-15
**Well written. Interesting approach.**

**Rating:** 6
**Confidence:** 4

**Review:**

In this paper, the authors incorporate principles from cortical connectivity at multiple scales to develop a transformer architecture that is more robust than the baseline transformer and contains lesser attentional parameters, while maintaining uncorrupted accuracy. In more detail, the authors use CCT as the baseline transformer and adapt its backbone to form the cortical transformer. They demonstrate that variants of the cortical transformer have better performance on the most difficult corruptions than the baseline transformer and transformers obtained by naively reducing the number of attentional parameters.

This paper is well-written. The adaptation of principles from cortical connectivity in this study is very interesting. However, there are a few weaknesses.
1. The biggest concern with the paper is that the variants of the cortical block show minor performance improvement over the baseline. Moreover, the improvement over the baseline is only observed for a small subset of the corruptions.
2. This is a minor point. The fact that the CCT-based cortical transformer shows only minor performance improvement raises a concern that this result may not generalize to other transformers. The study could have benefitted from adapting the cortical block to another transformer architecture to verify that the benefits of using cortical principles generalize.

---

> ### Author Response · Authors · 2022-12-28
> **Reviewer 1 Response**
>
> We appreciate the detailed review and contextualizing our results. The raised concerns are worth highlighting for readers of this work, however we feel that outcomes of this work are still helpful for the community and have added further clarification and contextualization in the paper:
>
> As noted in (1), while the performance is improved over the baseline, this performance increase is indeed relatively minor. We feel that this contribution is still important for the community considering that these networks have reduced learnable attention parameters by over an order of magnitude (even if the performance was unchanged or slightly decreased, but with fewer parameters this would still be important for highlighting strategies for network compression in biological and artificial systems). Also, while the improvements were not for all of the corruptions, the increases were for the most difficult corruptions. Again we feel this is worth highlighting to the research community, both A) As a strategy for practitioners that are the most concerned with increasing performance for highly corrupted inputs and B) For those studying the principles of neural processing - that certain architectural elements may be the most important for performing well in cases where network inputs are different than those used during network training.
>
> We have updated the discussion to highlight these points by adding "When all investigated principle are combined, even though the observed performance increases were modest, the total number of learnable attention parameters are reduced by a factor 15, which is a significant gain in model compression. Furthermore, the largest performance increases were observed for the most difficult natural corruptions, highlighting the potential utility of the investigated principles for domain shift in both biological and artificial systems" to the first paragraph.
>
> As noted in (2), we used a single CCT baseline architecture for comparing performance with the cortical variants.  This choice was motivated because the main difference in the CCT architecture over other vision transformers (e.g. Dosovitskiy et al. 2020) is in the input tokenizer and the sequence pooling output for classification, with the intermediate transformer-based backbone unchanged. The changes we investigate are modifications to the transformer-based backbone which are shared across vision transformers and a wide range of transformer based networks more broadly. Nevertheless, a more thorough analysis of the relationship between the properties of the input tokenizer and cortical properties would be informative.
>
> We update the first paragraph of the Experiments and Results section to make this clearer:
> "Importantly, the CCT network preserves a standard transformer-based backbone, obtaining parameter efficiency and performance gains through an updated image tokenizer before the transformer-based backbone and a sequence pooling operation for classification after the transformer-based backbone.  We leverage the convolutional tokenizer from this architecture and replace the transformer backbone with our Cortical Blocks to evaluate cortical design principles."